**Mineralogical response of the Mediterranean crustose coralline alga *Lithophyllum cabiochae* to near-**
**future ocean acidification and warming**
Merinda C. Nash[1,2*], Sophie Martin[3], Jean-Pierre Gattuso[4,5]
[1] Research School of Physics and Engineering, The Australian National University, Canberra, Australia
[2] Dept. of Botany, Smithsonian Institution, Washington DC, USA
[3]- Sorbonne Universités, UPMC Université Paris 06, UMR 7144, Station Biologique de Roscoff, 29680 Roscoff,
France
[4]- CNRS, UMR 7144, Laboratoire Adaptation et Diversité en Milieu Marin, Station Biologique de Roscoff,
29680 Roscoff, France
[5] Institute for Sustainable Development and International Relations, Sciences Po, 27 rue Saint Guillaume, F-
75007 Paris, France
Correspondence to: Merinda.nash@anu.edu.au




**Abstract**

Red calcareous coralline algae are thought to be among the organisms most vulnerable to ocean acidification due to the high solubility of their magnesium calcite skeleton. Although skeletal mineralogy is proposed to change as $CO_2$ and temperature continues rising, there is currently very little information available on the response of coralline algal carbonate mineralogy to near-future changes in $pCO_2$ and temperature. Here we present results from a one-year controlled laboratory experiment to test mineralogical responses to $pCO_2$ and temperature in the Mediterranean crustose coralline alga (CCA) *Lithophyllum cabiochae*. Our results show that Mg incorporation is mainly constrained by temperature (+1 mol% $MgCO_3$ for an increase of 3°C) and there was no response to $pCO_2$. This suggests that *L. cabiochae* thalli have the ability to buffer their calcifying medium against ocean acidification, thereby enabling them to continue to deposit Mg-calcite with a significant mol% $MgCO_3$ under elevated $pCO_2$. Analyses of CCA dissolution chips showed a decrease in Mg content after 1 year for all treatments but this was not affected by $pCO_2$ nor by temperature. Our findings suggest that biological processes exert a strong control on calcification on Mg-calcite and that CCA may be more resilient under rising $CO_2$ than previously thought. However, previously demonstrated increased skeletal dissolution with ocean acidification will still have major consequences for the stability and maintenance of Mediterranean coralligenous habitats.

**Keywords:** Ocean acidification, carbonate skeleton, coralline algae, global warming, mineralization, Mg-calcite, $CO_2$, temperature

## 1. INTRODUCTION

Coralline algae are thought to be among the organisms most vulnerable to ocean acidification (decreasing pH and increasing $p$CO$_2$). This is because their skeletons consist of magnesium-calcite (Mg-calcite) and the solubility of Mg-calcite (> 8-12 mol% MgCO$_3$) is greater than the solubility of the other forms of calcium carbonate (CaCO$_3$), calcite (low Mg-calcite, <8 mol%) and aragonite (Andersson et al., 2008). Consequently, it has been suggested that coralline algae, both articulated and crustose coralline algae (CCA), will be among the first organisms to dissolve in the context of ocean acidification (Andersson et al., 2008). However, the presence of Mg-calcite phases with lower solubility such as dolomite (50 mol% MgCO$_3$) within the cells of tropical CCA, results in reduced dissolution rates (Kline et al., 2012; Nash et al., 2013a). Potential resilience of coralline algae to ocean acidification may thus occur through changes in skeletal mineralogy either by producing calcite with lower Mg content (Chave, 1954; Agegian, 1985; Stanley et al., 2002; Ries, 2011; Egilsdottir et al., 2013) or by favoring accumulation of CaCO$_3$ forms with lower solubility such as dolomite (Diaz-Pulido et al., 2014). The Mg content in coralline algae is also known to vary as a function of seawater temperature (Agegian, 1985; Halfar et al., 2000; Kamenos et al., 2008; Hetzinger et al., 2009; Caragnano et al., 2014; Diaz-Pulido et al., 2014), which is considered to exert a primary control by facilitating Mg incorporation into the skeleton (Kamenos et al., 2008). However, there is currently limited information available on the response of the mineralogy of coralline algae to near-future changes of $p$CO$_2$ and temperature, and none on temperate CCA.

The response of dead CCA crust to varying pH conditions is also of interest as it is the preservation of this crust that underpins many of the coralligenous habitats. It has been proposed that as CO$_2$ rises, calcite with higher Mg-contents will dissolve and may re-precipitate as lower Mg-phases (Andersson et al., 2008). This would result in lower average mol% MgCO$_3$ of Mg-calcite. As it is proposed than calcite with lower Mg-contents are more thermodynamically stable that those with higher Mg-content, that could provide a positive feedback mechanism to stabilize the calcium carbonate skeletons. As yet, there has been no experimental work on the Mg-calcite skeletons of CCA to test this proposal. An experiment performed on skeletal chips of *Lithophyllum cabiochae* reported rates of dissolution 2 to 4 times higher under elevated $p$CO$_2$ than under ambient $p$CO$_2$ (Martin and Gattuso 2009). These dissolution chip samples offer an opportunity to test the theory that higher Mg phases of Mg-calcite would dissolve differentially from the CCA crusts potentially increasing the stability of the dead substrate.

We investigated experimentally the response of the carbonate mineralogy of the CCA *Lithophyllum cabiochae*,
one of the main calcareous components of coralligenous habitats in the Mediterranean Sea, after 12 months of
exposure to ocean acidification and warming. The hypotheses tested are: (1) the Mg content of the new growth
would increase with temperature, (2) the Mg content of the new growth would decrease under elevated $pCO_2$,
and (3) the Mg content of dead dissolution chips would decrease with elevated $pCO_2$.


**2. MATERIALS AND METHODS**
Full experimental details, carbonate chemistry, growth, respiration, photosynthesis, net calcification and
dissolution rates can be found in Martin and Gattuso (2009) and Martin et al. (2013a). A summary follows.
Specimens of the CCA *Lithophyllum cabiochae* (Boudouresque & Verlaque) Athanasiadis were collected in the
coralligenous community at *ca.* 25 m depth in the Bay of Villefranche (NW Mediterranean Sea, France;
43°40.73'N, 07°19.39'E) on 10 July 2006 and transported to the laboratory in thermostated tanks within 1 h. Flat
thalli were selected for the experiments and were thoroughly cleaned of epiphytic organisms. They were
randomly assigned in four 26-L aquaria and reared for one year (July 2006-August 2007) in four treatments:
(1) ambient $pCO_2$ (*ca.* 400 µatm) and ambient temperature (*T, i.e.* the temperature at 25 m depth in the Bay of
Villefranche; control, labeled *400 T*),
(2) ambient $pCO_2$ and elevated temperature (*T+3*°C; *400 T+3*),
(3) elevated $pCO_2$ (*ca.* 700 µatm) and ambient temperature (*700 T*),
(4) elevated $pCO_2$ and elevated temperature (*700 T+3*).
A further set of CCA thalli were air dried until dead and placed in the tanks in December 2006 for the remaining
8 months of the experimental period to measure rates of dissolution  (Martin and Gattuso, 2009). The aquaria
were continuously supplied with Mediterranean seawater from two 110-L header tanks in which $pCO_2$ was
adjusted by bubbling ambient air (ambient $pCO_2$) or $CO_2$-enriched air (elevated $pCO_2$) obtained by mixing pure
$CO_2$ to ambient air.  Temperature was gradually changed according to the season from *T* = 13.3 to 22.0°C (*T+3*
= 16.3 to 25.0°C). Irradiance was set to the mean *in situ* daily irradiance at 25 m depth in the Bay of
Villefranche and was adjusted seasonally from 6 to 35 µmol photons m$^{-2}$ s$^{-1}$. The photoperiod was adjusted
weekly according to natural fluctuations and varied from 9:15 (Light:Dark ratio - winter) to 15:9 (summer). The
annual means of the carbonate chemistry parameters are shown in Table 1.At the end of the experiment all crusts
were air-dried. Four sets of crust were sampled for    X-ray diffraction (XRD): (1) the new crusts grown from the
bottom face of the main thalli (Figure 1), (2) the pink surficial crust on the original thalli, (3) the original thalli
(Fig. 1) and (4) pieces of dead crust that had been used for dissolution tests.  New crusts were confirmed to have
grown during the experiment as the pre-existing crust was cleaned and photographed at the time of collection
and these growths were not present at that time. For the new crusts, sets of 4-5 crust fragments similar in size
(*ca*. 2-3 mm in diameter) and thickness (~ 1 mm thick), were randomly selected from 8 thalli per treatment. To
obtain sufficient material for XRD analyses of the new thalli, 3-4 crust fragments were used from each alga.
For the original thalli and dissolution chips, subsamples ~2-3 mm thick were cut off the sides. The pink surface
of the original thalli was sampled by gently scraping with a razor ensuring not to scrape into the white crust
underneath.  This uppermost surface was presumed to have grown during the experiment and sampled instead of
the surfaces of the new crusts as there was not a large enough surface area on the new protrusions to collect
sufficient pink crust for analyses. Scrapings from 5 algae from each treatment were required in order to obtain
enough material for one XRD test.  The depth of the pink-pigmented crust was ~200- 500 μm but only the
surface is sampled by scraping on the top with the aim of collecting predominantly epithallial material.
However, we do not refer to it as epithallus because by this sampling method we cannot confirm that no sub-
perithallial crust has been included, hence surficial pink crust is the most accurate description of this subsample.
Our development of this method has shown that if too much pressure is applied during the scraping then
substantial amounts of perithallial crust, that also can be pink-pigmented, may be unintentionally sampled.
The mol% $MgCO_3$ of the crust fragments were determined via XRD using a Siemens D501 Bragg-Brentano
diffractometer equipped with a graphite monochromator and scintillation detector, using Cu*Kα* radiation. Crust
fragments were crushed and powdered with fluorite ($CaF_2$) added as an internal standard. The coralline crust and
fluorite are ground together with a mortar and pestle. Enough fluorite was added to obtain a clear peak, this is
usually between 5-20 weight percent but not specifically weighed.  Powdered samples were mounted on a low
background quartz slide. For the pink surficial crust, more fluorite (30-50 %) was added to obtain enough
powder to cover the centre of the quartz slide. Mg-content of calcite was calculated from the (104) peak position
and any asymmetry present was quantified as described in Nash et al. (2013b).  XRD scans with 25-32° 2-theta
scan length were processed using EVA Diffract Plus software packages and interpreted following procedures
described Nash et al. (2013b) and further developed in Diaz-Pulido et al. (2014). XRD measurements had a
reproducibility of ± 0.11 mol% (standard deviation; *n*=3 analytical repeats of sample 700T+3, 5a).
The effects of $p\text{CO}_2$ and temperature were assessed by two-way ANOVAs and followed by Tukey HSD post hoc
tests. Normality of the data and homoscedasticity were checked by Kolmogorov-Smirnov's test and Levene's
test, respectively. A t-test was completed to compare asymmetry differences between the main thalli and
dissolution chips.

**3. RESULTS**
In general, the Mg content of the CCA's increased with temperature but was not affected by $\text{CO}_2$ (Fig. 2).
Dissolution chips had lower Mg content than the main thalli and neither the main thalli (pre-experimental crust)
or the dissolution chips showed any trends with temperature or $\text{CO}_2$ (Table 2).
**3.1 New crust**- XRD results indicate that the new crusts of *L. cabiochae* are entirely calcitic (Mg-calcite). The
mean (±standard deviation) Mg contents were $15.2 \pm 0.7$, $16.0 \pm 0.5$, $15.0 \pm 0.5$, and $16.1 \pm 0.3$ mol% $\text{MgCO}_3$ in
the *400 T*, *400 T+3*, *700 T*, and *700 T+3* treatments, respectively (Fig. 2, the complete data set is provided in
Supplementary information table 1). The Mg-calcite peaks were symmetrical indicating there was no dolomite,
nor magnesite present. The Mg content was significantly affected by temperature (2-way ANOVA, $p < 0.0001$),
being about 1 mol% $\text{MgCO}_3$ higher at elevated temperature (+3°C) relative to ambient temperature at both $p\text{CO}_2$
levels but was not affected by $p\text{CO}_2$ (Fig. 2; Table 2A).
**3.2 Pink surficial crusts**- The pink surficial crusts were also entirely Mg-calcite. The Mg content was 14.3,
14.6, 14.6 and 15 mol% $\text{MgCO}_3$ in the *400 T*, *400 T+3*, *700 T*, and *700 T+3* treatments, respectively (Fig. 2).
There is no standard deviation or statistical analysis of the pink surficial crust results because only one analysis
was performed on material from 5 thalli combined for each treatment.
**3.3 Main thalli**- The mean (± standard deviation) Mg content in the main thalli were $16.0 \pm 0.5$, $16.1 \pm 0.4$, $15.6$
$\pm 0.4$, and $16.1 \pm 0.6$ mol% $\text{MgCO}_3$ in the *400 T*, *400 T+3*, *700 T*, and *700 T+3* treatments, respectively. The
Mg content was not affected by temperature or $p\text{CO}_2$ (Table 2B, SI. Table 2). There was minor asymmetry on
the higher mol% $\text{MgCO}_3$ side of Mg-calcite XRD peaks indicating the presence of a higher Mg-calcite phase
(Fig. 3). However, this asymmetry did not extend over the dolomite position suggesting the extra phase was a
second Mg-calcite. The difference in mol% $\text{MgCO}_3$ when incorporating the extra asymmetry into the
calculations (see Nash et al. 2013b for full discussion on this method) showed that the asymmetry was also not
affected by temperature or $p\text{CO}_2$ (Table 3).
**3.4 Dissolution chips**- The mean (± standard deviation) Mg content of dissolution chips were 15.4 ± 0.5, 15.6 ±
0.5, 15.6 ± 0.5, and 15.5 ± 0.4 mol% $MgCO_3$ in the *400 T*, *400 T+3*, *700 T*, and *700 T+3* treatments,
respectively. The Mg content was not affected by temperature or $pCO_2$ (Table 2C, SI Table 3). The average Mg
content was significantly lower in the dissolution chips than in main thalli (15.5 ± 0.4 versus 16.0 ± 0.5 mol%
$MgCO_3$, t-test, p < 0.001) (Fig. 3 A, B). Similarly to the main thalli, there was a minor asymmetry on the higher
mol% $MgCO_3$ side of the Mg-calcite XRD peak indicating a second phase of Mg-calcite with higher Mg content
(Fig. 3 B). The difference in asymmetry was lower for the dissolution chips than the main thalli (t-test, p =
0.008; Fig. 3 C) and was not affected by temperature or $pCO_2$ (Table 3).

**4. DISCUSSION**
Results obtained on the new crust demonstrate that the mineralogy of *L. cabiochae* is primarily controlled by
temperature and scarcely constrained by $pCO_2$. Similarly, the Mg content does not respond to $pCO_2$ in dead
CCA skeleton but decreases in all dead crusts over the 12-month experiment. Thus our hypothesis that the Mg
content would increase with temperature is supported but the hypothesis that Mg content would decrease with
$pCO_2$ is not. Seawater temperature is effectively considered to exert primary control on Mg content in coralline
algae (Halfar et al., 2000; Kamenos et al., 2008). In *L. cabiochae*, an increase of 3°C above ambient temperature
led to an increase in Mg incorporation of 1 mol% $MgCO_3$, (0.33 mol% $MgCO_3$/°C) which is consistent with the
values reported in the literature, both experimentally and *in situ,* ranging between 0.3 and 2 mol% $MgCO_3$ per
°C (Chave and Wheeler, 1965; Halfar et al., 2000; Kamenos et al., 2008; Hetzinger et al., 2009; Caragnano et al.,
2014; Diaz-Pulido et al., 2014; Williamson et al., 2014). Conversely, $pCO_2$ did not drive significant change in
the Mg content of living *L. cabiochae*.

The Mg content of the pink surficial crust was higher in the elevated temperature treatments and while no
statistical analyses could be carried out, these results are consistent with the increase in Mg measured for pink
surficial crust as a function of increasing temperature reported in previous work (Diaz-Pulido et al., 2014). The
lower Mg content recorded for the pink surficial crust relative to the bulk crust is in agreement with previous
studies on CCA *Porolithon onkodes* (Diaz-Pulido et al., 2014; Nash et al., 2015). Sampling of the surface aims
to capture predominantly epithallial carbonate. The epithallial cells of corallines are typically different in shape
to the perithallial cells, being shorter and flattened or ovoid shape (e.g. Pueschel and Keats 1997). It is not
presently known why the epithallus has a lower Mg than the bulk perithallus or if this offset is common to all
species. However, the close agreement of the temperature response for the *P. onkodes* surficial crust of 0.37
mol% $MgCO_3$/°C (Diaz-Pulido et al. 2014) with the 0.33 mol% $MgCO_3$/°C measured for the new crust in this
experiment does suggest the controls on Mg uptake are similar for both the perithallial and epithallial cells.

The consistent shift of ~0.33 mol% $MgCO_3$/°C across both the control and $CO_2$ treatments, consider together
with the results for Diaz-Pulido et al. (2014) of 0.37 mol% $MgCO_3$/°C from a 2°C increase suggests the
magnesium change is a robust temperature response and this is of interest for CCA paleo temperature proxies
(e.g. Halfar et al. 2000, Kamenos et al. 2008, Hetzinger et al., 2009). This  increase in Mg content is also in
agreement with results obtained by XRD of articulate corallines [0.286 – 0.479 mol% $MgCO_3$/°C  (Williamson
et al. 2014)] and a variety of species [0.36 mol% $MgCO_3$/°C using only XRD results in Chave (1954)] collected
across a geographical temperature range.  The reports of ratios of up to 2 mol% $MgCO_3$/°C (Halfar et al., 2000:
Kamenos et al., 2008, Hetzinger et al., 2009; Caragnano et al., 2014; Williams et al. 2014) may be due to
different analytical methods or species-specific effects. The similarity of the results obtained using XRD for both
experimental and *in situ* corallines supports using a ratio of ~0.3 to 0.4 mol% $MgCO_3$/°C as a paleo thermometer
when the analytical methods return an effective spatial average for Mg-calcite and the absence of other
carbonates; aragonite, low-Mg-calcite, dolomite and magnesite, has been confirmed.

The new crusts represent the average mol% $MgCO_3$ influenced by the temperature experienced during their
growth, which is unlikely to be an even representation of the entire experimental duration. This is for two
reasons. First, growth rate varies with temperature which could result in a bias towards warmer months Mg-
content.  Secondly, although the progression of the new crust growth was not specifically monitored throughout
the experiment, it is likely that a relatively small amount of the final material would have formed during these
first few months. For the new growths to form, initially new hypothallial cells would have to bud out and then
change to perithallial cells, which form the bulk of CCA crust material.  The experiment started and finished in
summer, July.  The warmest months were July, August and September (Martin et al. 2009).  Calcification rates
during autumn, winter, spring ranged from ~0.02 – 0.15 $\mu$mol $CaCO_3$ cm$^{-2}$h$^{-1}$, in contrast to the substantially
higher rates of 0.3 – 0.52 $\mu$mol $CaCO_3$ cm$^{-2}$h$^{-1}$ during summer months (Martin et al. 2013). While growth may
have commenced during the summer, probably most of the crust formed over the remainder of the experimental
period, autumn, winter and spring. However, that there was no significant difference in mineralogy between the
treatments except for the temperature influence, suggests that all crusts analysed grew over similar time frames,
i.e. each crust has similar proportions of growth from the differing temperature and time periods.

Prior to utilizing CCA as a temperature proxy, it is necessary to verify that the Mg in the Mg calcite is the
primary Mg incorporated during calcification and not the result of diagenesis. The depletion of Mg in the
dissolution experiment crusts over eight months indicates this change can be relatively rapid once the organism
is no longer protected by living tissue. This is likely to be more of a problem when using fossil branching
corallines than thick crusts that retain a living surface layer. Indeed, Kamenos et al. (2008) noted their sub-fossil
*Lithothamnion glaciale* had significantly lower Mg in the summer season than their living samples. The
likelihood of remineralization was considered by Kamenos et al. but rejected, as remineralization was presumed
to be to either low Mg calcite or aragonite. The possibility of remineralization to a lower phase of Mg-calcite, as
occurred in this experiment and noted at the base of CCA *Porolithon onkodes* (Nash et al. 2013b) had not
previously been reported. The present study experimentally confirms that diagenesis does not necessarily mean a
change to aragonite or low Mg-calcite, but instead can be to a lower phase of Mg-calcite thus making it more
difficult to detect post-mortem diagenesis from Mg measurements or mineralogy alone.  High magnification
SEM work would be required to check for remineralization.

Analyses of the pre-existing thalli (main thalli) provide a baseline Mg content for pre-experimental *L. cabiochae*
with the assumption that this has not changed during the experiment. The average across treatments was 16.1
mol% $MgCO_3$, excluding the 700T treatment.  This Mg content is higher than that of the new crusts grown under
ambient temperatures (400T and 700T) and is probably due to a larger amount of pink surficial crust with lower
Mg content in the thin new crusts relative to the pre-existing thicker thalli. Although the lower average for 700T
is not significantly different from the other three treatments, when this lower Mg content is considered in the
context of the results for the dissolution chips the lower measurement takes on greater relevance. Results for the
dissolution chips were not significantly different between treatments with a combined average of 15.5 mol%
$MgCO_3$.  This was significantly lower than the pre-existing thalli for all treatments except the 700T, suggesting
the 700T main thalli may have undergone alteration similarly to the dissolution chips during the experiment.
The values for 700T were compared by t test to the combined dataset for 400T, 400T+3 and 700T+3 and were
significantly lower than the group, $p = 0.0531$.

Ideally when carrying out experiments where it is planned to analyse the crust, for either mineral or structural
changes, it is best if a subsample is taken from of each piece prior to being placed into the experimental tanks.
This way it can be established that post-experiment crust features are truly representative of the environmental
sample (e.g. Nash et al. 2013a) and have not been altered by virtue of being placed in tanks for the duration as is
suspected for 700T.  Problematically some CCA exhibit changes in growth unrelated to treatment after being
placed in tanks (Nash et al. 2015). Thus best practice would be to keep aside subsamples, particularly where a
species has not already been well studied at the cellular scale, so that it can be determined if the control tanks
result in growth and mineral composition comparable to in-situ growth.

It is interesting to consider why the dissolution chips have lower Mg content than the main thalli when they were
subsamples of the same. Presumably, because the thalli remained covered in living tissue, this has substantially
protected the crust from exposure to ambient seawater whereas the dissolution chips had direct exposure to
seawater. Assuming that the dissolution chips initially had the same Mg content as the main thalli from which
they were subsampled, then, the lower Mg content after 8 months of direct exposure to seawater indicates there
has been alteration of the crust.  All chips lost weight over the 8 months (Martin and Gattuso 2009) with those in
the 700T and 700T+3 treatments having the highest dissolution rates. However, the absence of a trend for Mg
content with treatment indicates that dissolution rates do not influence the thermodynamics of the Mg-calcite
dissolution process for these CCA. The strong correlation with $CO_2$ and dissolution rates in the original
experiment (Martin et al. 2009) suggests that the seawater pH is the dominant factor in driving dissolution.
Indeed micro-bioerosion can increase in response to that, as shown in Diaz-Pulido et al. (2014), but the
mechanism by which they could influence the Mg-content of the Mg-calcite is not yet known.  Further work is
planned to analyse these dissolution chips by SEM-EDS which may shed light on the process.

Theory suggests that the higher phases of Mg-calcite will dissolve first (Andersson et al., 2008) but micro-
structural properties may interfere with a purely thermodynamic response (Morse et al., 2007; Henrich and
Wefer, 1986; Walter and Morse, 1985, reviewed in Eyre et al., 2014; Pickett and Andersson, 2015). The lowest
phase in the *L. cabiochae* is the pink surficial crusts but they do not make up a substantial amount of the main
thalli bulk sample. The presence of asymmetry indicates an extra phase of Mg-calcite with a higher content of
Mg. Previous works on cold water (Adey et al., 2014) and tropical (Nash et al., 2013a) CCA have shown that the
cell wall and inter-filament regions have visually different crystal morphology. It may be that they have different
Mg content although this hypothesis has not been tested yet. Statistical results showed lower asymmetry for the
dissolution chips compared to the main thallus. This indicates that the relative proportion of higher-Mg-phase
Mg-calcite was less in the dissolution chips suggesting that the higher-Mg-phase, while still present, had
suffered greater dissolution relative to the lower-Mg-phases. Dissolution experiments have demonstrated that the
inter-filament Mg-calcite is the first to dissolve in pH 8 (NBS) after 1 h (Nash et al., 2013) and the cell walls
remain intact until exposed to pH 7.7-7.82 over several hours. The pH in the present experiment did not drop
below $pH_T$ 7.8 in the 700T or 700T+3 treatments (Martin and Gattuso, 2009). Considering these previous
studies and the data presented here, it seems likely that the cell walls have remained substantially intact but the
inter-filament Mg-calcite has remineralized to a lower phase of Mg-calcite and there may also be abiotic Mg-
calcite infilling cell spaces prior to complete dissolution of the exposed edge. The process of cell infill by Mg-
calcite has been observed in the exposed bases of tropical CCA *P. onkodes* (Nash et al., 2013a) whereby exposed
dead cells are in-filled with Mg-C. XRD analyses of the exposed base of the tropical CCA measured 14.8 mol%
$MgCO_3$ compared to the main crust of 16.9 mol% $MgCO_3$ (Nash et al., 2013b) indicating that the abiotic Mg-
calcite has lower average Mg content than the original crust.

If the proposal for remineralization of the dissolution chips is correct, then the results for the present study would
indicate that there is no trend with Mg and temperature or $CO_2$ for abiotic mineral formation. This would be in
contrast to results for synthetic formation of Mg-calcite (Mucci 1987) although the trend for synthetic Mg
content was substantially less sensitive than uptake for biogenic Mg-calcite, with an increase of only 2 mol%
$MgCO_3$ from 6 to 8 mol% $MgCO_3$ from 5 to 25°C. Support is provided for the absence of temperature trend by
another comparison of the results for the dissolution chips to dead tropical CCA sampled from a coral reef core
from Rodrigues Island, Indian Ocean (Rees et al., 2005) where the Mg content of the dead crusts was 15 to 15.3
mol% $MgCO_3$ (Nash et al., 2013b). To thoroughly test the hypothesis for an absence of temperature trend in
abiotic Mg-calcite mineralization, a comprehensive survey of dead CCA from a range of latitudes would be
required. However, the clear trend for increase in Mg uptake by living CCA as temperature increases, compared
to the absence of trend in altered dissolution chips, suggests the Mg content increase may be primarily driven by
a biological response, rather than abiotic thermodynamics alone that the organism is unable to compensate for as
suggested by Diaz-Pulido et al. (2014).

Although earlier studies on Mg incorporation in the skeleton of coralline algae grown experimentally have found
a decline in Mg content with higher $pCO_2$, likely conferring them a better resistance to dissolution (Agegian,
1985; Ries, 2011; Egilsdottir et al., 2013; Ragazzola et al., 2016), the lack of a $pCO_2$ effect in *L. cabiochae* is
consistent with recent findings (Kamenos et al., 2013; Diaz-Pulido et al., 2014; Nash et al., 2015) suggesting that
skeletal mineralogy may be under biological control. The organic substrate may template a baseline magnesium
proportion, which then only changes in response to temperature. The ability of coralline algae to control the
carbonate chemistry (pH/$pCO_2$ and carbonate saturation state) of the calcifying medium through metabolic
activities could enable them to continue to deposit Mg-calcite with a relatively high mol% $MgCO_3$ despite
changes in the carbonate chemistry driven by ocean acidification (Kamenos et al., 2013, Diaz-Pulido et al., 2014)
as has been inferred for other Mg-calcite organisms (Ries 2011). A biological control of mineralization by
coralline algae has already been inferred in *L. cabiochae* because its rate of calcification is maintained or even
enhanced under elevated $pCO_2$ (Martin et al., 2013a).

It is worth considering whether there is a compensatory mechanism enabling Mg-content maintenance in the
elevated pCO₂ treatment. This consideration implies that the Mg content automatically declines with lower pH
(the hypothesis we tested) and the organism must therefore have compensated because the results showed no
difference with pCO₂. While there have been many studies on Mg content responses to elevated $pCO_2$ treatments
(e.g. Ries, 2011; Egilsdottir et al., 2013; Kamenos et al., 2013; Diaz-Pulido et al., 2014; Nash et al., 2015;
Ragazzola et al., 2016) as yet, there has been no study on the internal cellular-scale metabolic controls on Mg
uptake in coralline algae. That is, the controls of Mg uptake are unknown.  Is it the internal carbonate chemistry,
the type of organic substrate or a combination of both? Without an understanding of the physiological
mechanisms that control Mg uptake, it is impossible to do more than speculate about potential compensatory
metabolic processes. This inhibition to carrying out an informed analysis of potential metabolic controls on Mg
uptake highlights the need for basic scientific investigation into how coralline algae calcify and what role the
anatomy and organic substrates play in calcification and Mg uptake. Then, we could start to understand how
these processes react to external environmental changes. Ragazzola et al. (2013, 2016) and Hoffman et al. (2012)
have shown anatomical changes in response to $pCO_2$ that may be ameliorated over longer time periods but the
exact controls on those changes are not known. Recent work has demonstrated the capacity of CCA to maintain
elevated pH in the boundary layer (~ 100 microns thick, dependent on water motion) when ambient seawater pH
is reduced (Hofmann et al. 2016).  The maintenance of the Mg content, if it is related to pH or saturation state,
may be enabled by the organisms capacity to control boundary layer pH and thus effectively inhibit the treatment
pH from reaching the living surface.

It remains unclear to what extent the algal metabolism exerts a control on Mg-carbonate chemistry as different
effects of $p$CO$_2$ on the Mg content and calcification rates have been found in other species of coralline algae
(Ries, 2011; Ragazzola et al. 2013, 2016). The increase in Mg content at elevated temperature may lead to
increased thalli dissolution but this could be offset by increased calcification (Martin et al., 2013a). However, the
enhanced mortality under the combination of projected ocean warming and acidification (Martin and Gattuso,
2009) could have major consequences for the physical stability and maintenance of coralligenous habitats that
outweigh any adaptive mineral response. Further work to understand the process that leads to lower Mg content
in the dead algal chips post mortem would shed light on remineralization of CCA post-mortem.

**Author contributions**
S.M. and J.P.G conceived and carried out the experimental work. M.N. carried out the mineral analyses. All
authors contributed to writing the MS.

**Data availability**
All raw data used for statistical analyses is included in the supplementary information.

**ACKNOWLEDGEMENTS**
This work was supported by the CarboOcean IP of the European Commission (grant 511176-2) and is a
contribution to the "European Project on Ocean Acidification" (EPOCA) which received funding from the
European Community (grant agreement 211384).

Authors declare no existing competing financial interests in this work.

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

**Figures**

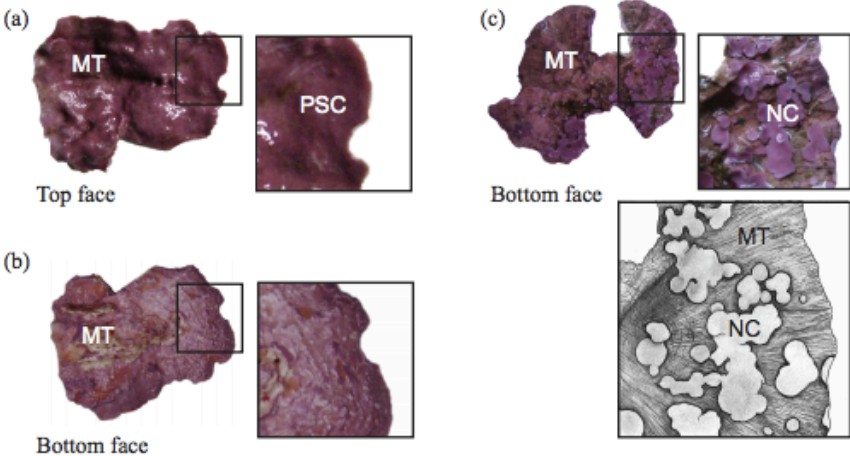


Figure 1: (a) Top face of the main thallus (MT) of *L. cabiochae* showing the pink surficial crust (PSC) and
bottom faces (b) free of crusts at the time of collection and (c) with new crusts grown during the experimental
period (photos and drawing S. Martin)

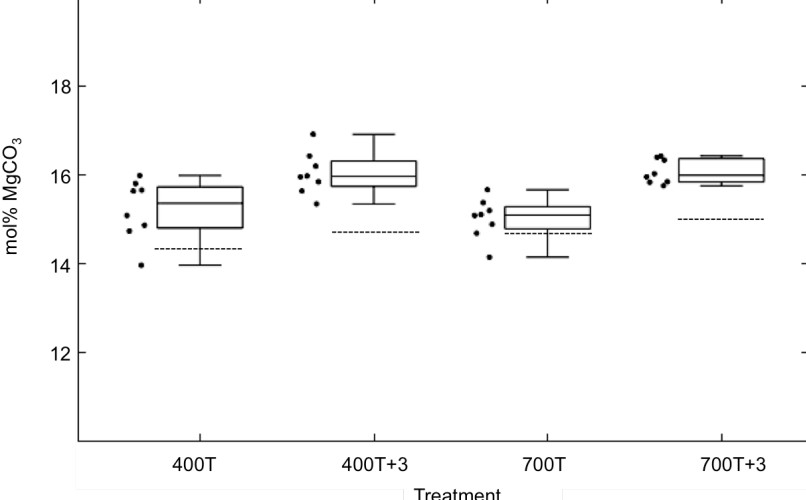


**Figure 2**: XRD results for new crust and pink surficial growth from the 400 and 700 $\mu$atm treatments, in ambient
temperature and ambient + 3 °C. Dashed lines shown the mean mol% $MgCO_3$ for pink surficial growth. The box
plots represent the new crust and the dots are individual data points. The boxes represent 25 and 75 percentiles,
the horizontal bold line is the median value and the whiskers are minimum and maximum values.



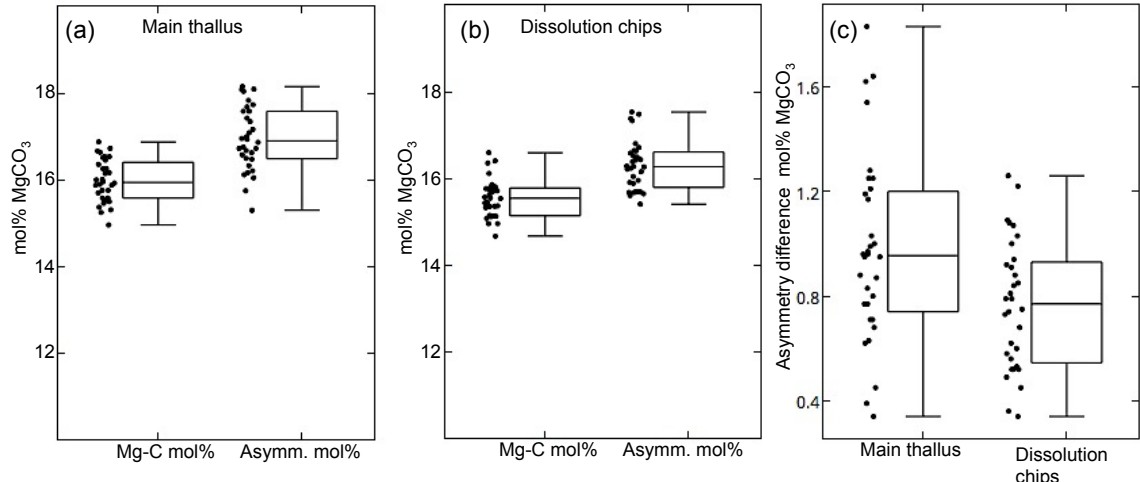


**Figure 1**: XRD results for main thalli and dissolution chips. **(a)**. Mol% $MgCO_3$ and asymmetry mol% $MgCO_3$
for the main thallus. **(b)**. Mol% $MgCO_3$ and asymmetry mol% $MgCO_3$ for the dissolution chips. **(c)**. Difference
in asymmetry mol% $MgCO_3$ between the main thallus and dissolution chips indicating a reduction in the higher
phases of Mg-calcite after dissolution.

 **Table 1** Parameters of the carbonate system in each treatment.

The values reported are means (± standard error) of 191 to 194 data collected from July 2006 to August
2007. The pH ($pH_T$, on the total scale) and total alkalinity ($A_T$) were measured while other parameters were
calculated. $pCO_2$, $CO_2$ partial pressure; $C_T$, dissolved inorganic carbon; $\Omega_c$ and $\Omega_a$, saturation state of
seawater with respect to calcite and aragonite.

| Treatment | $pH_T$ (total scale) | $A_T$ (mmol kg$^{-1}$) | $pCO_2$ (uatm) | $CO_2$ (mmol kg$^{-1}$) | $CO_3^{2-}$ (mmol kg$^{-1}$) | $HCO_3^-$ (mmol kg$^{-1}$) | $C_T$ (mmol kg$^{-1}$) | $\Omega_c$ | $\Omega_a$ |
|---|---|---|---|---|---|---|---|---|---|
| 400 T | 8.08 ± 0.00 | 2.516 ± 0.004 | 397 ± 2 | 0.014 ± 0.000 | 0.226 ± 0.001 | 1.974 ± 0.003 | 2.213 ± 0.002 | 5.26 ± 0.03 | 3.41 ± 0.02 |
| 400 T+3 | 8.05 ± 0.00 | 2.519 ± 0.004 | 436 ± 3 | 0.014 ± 0.000 | 0.233 ± 0.001 | 1.962 ± 0.004 | 2.208 ± 0.002 | 5.43 ± 0.03 | 3.55 ± 0.02 |
| 700 T | 7.87 ± 0.00 | 2.517 ± 0.004 | 703 ± 3 | 0.024 ± 0.000 | 0.152 ± 0.001 | 2.155 ± 0.003 | 2.331 ± 0.002 | 3.54 ± 0.03 | 2.30 ± 0.02 |
| 700 T+3 | 7.85 ± 0.00 | 2.523 ± 0.004 | 753 ± 3 | 0.024 ± 0.000 | 0.159 ± 0.001 | 2.144 ± 0.004 | 2.326 ± 0.003 | 3.72 ± 0.03 | 2.43 ± 0.02 |





**Table 2** ANOVA testing the effect of $pCO_2$ and temperature on skeletal mol% $MgCO_3$ in (A) new crusts,
(B) main thalli, and (C) dissolution chips of *Lithophyllum cabiochae.*

| Source | df | MS | $F$ | p |
|---|---|---|---|---|
| **A) New crusts** | | | | |

| Source | df | MS | F | p |
|---|---|---|---|---|
| $p\mathrm{CO_2}$ | 1 | 0.000005 | 0.223 | 0.65 |
| Temperature | 1 | 0.000701 | 28.620 | **<0.0001** |
| $p\mathrm{CO_2} \times$ temperature | 1 | 0.000011 | 0.444 | 0.51 |
| Error | 28 | 0.000024 | | |
| **B) Main thalli** | | | | |
| $p\mathrm{CO_2}$ | 1 | 0.000014 | 0.601 | 0.44 |
| Temperature | 1 | 0.000048 | 2.094 | 0.16 |
| $p\mathrm{CO_2} \times$ temperature | 1 | 0.000042 | 1.844 | 0.19 |
| Error | 28 | 0.000023 | | |
| **C) Dissolution chips** | | | | |
| $p\mathrm{CO_2}$ | 1 | 0.000003 | 0.143 | 0.71 |
| Temperature | 1 | 0.000005 | 0.218 | 0.64 |
| $p\mathrm{CO_2} \times$ temperature | 1 | 0.000014 | 0.663 | 0.42 |
| Error | 28 | 0.000021 | | |





**Table 3** ANOVA testing the effect of $p\mathrm{CO_2}$ and temperature on difference in asymmetry mol% $MgCO_3$ in
(A) main thalli and (B) dissolution chips of *Lithophyllum cabiochae.*

| Source | df | MS | F | p |
|---|---|---|---|---|
| **A) Main thalli** | | | | |
| $p\mathrm{CO_2}$ | 1 | 0.000008 | 0.569 | 0.46 |
| temperature | 1 | 0.000006 | 0.441 | 0.51 |
| $p\mathrm{CO_2} \times$ temperature | 1 | 0.000007 | 0.489 | 0.49 |
| Error | 28 | 0.000013 | | |

**B) Dissolution chips**

| | | | | |
|---|---|---|---|---|
| $p$CO$_2$ | 1 | 0.000022 | 3.871 | 0.06 |
| temperature | 1 | 0.000001 | 0.190 | 0.67 |
| $p$CO$_2$ × temperature | 1 | 0.000005 | 0.944 | 0.34 |
| Error | 28 | 0.000006 | | |
