# Peer review of "Mineralogical response of Mediterranean crustose coralline algae *Lithophyllum cabiochae* to near-future ocean acidification and warming"

_Biogeosciences, 2016_

## Referee Comment (RC1) · B. Steneck (Referee) · 31 May 2016

This is an interesting study of the mineralogical responses of a Mediterranean coralline alga under elevated levels of pCO2 and temperature. The study compared live and dead portions of the crust to sort out abiotic effects from those that are biologically mediated on how the alga incorporates Mg into its MgCaCO3 thallus.

Interestingly, there was no change in the % MgCO3 under elevated pCO2. Because the study duration was a year, seasonal changes were integrated. Although the paper is not an easy one to read because it is so information dense, it is important and should

have a reasonable impact beyond just the coralline algal specialists. There are more papers in recent years indicating that greenhouse gas impacts on calcifying organisms is more complicated than we once thought. How organisms respond metabolically to elevated levels of carbon dioxide requires more study. This is a solid piece of work that only scratches the surface. For example, what evidence is there of the metabolic costs of mediating elevated levels of carbon dioxide? If this is metabolically expensive, might other factors suffer from these costs? For this coralline, how does overall growth rate change as a function of temperature and pCO2? Those questions are beyond the scope of the present study but they do suggest new avenues of research this sort of study will likely stimulate.

Specific comments:

This is generally well written.

The pigmented zone on many corallines does not delimit the living tissue. However, on the thin pieces illustrated in Fig. 1 they may. Nevertheless, the "pink" thallus simply has pigments and it can vary in depth into the thallus. I did not see anywhere, where the thickness of the pigmented tissue was reported for the upper and for the lower surface of these crusts. Given the attention given on mineralization that occurs in the pigmented region, that thickness seems to me to be important. (e.g., see lines 200 – 202)

Line 71: replace "preferentially with differentially.

Line 108: Fragments 2 – 3 mm are extremely small with considerable exposed thallus relative to the entire photosynthetic surface area. Could this have had an effect on the outcome of the experiment?

Line 108: I suggest you replace the second use of the word "diameter" with "thickness".

---

## Referee Comment (RC2) · Anonymous Referee #2 · 22 Jun 2016

The manuscript of Nash et al. reports interesting results of a study combining culturing (under varying T and pCO2 conditions) and dissolution experiments on dead thalli of CCA species L. cabiochae. It is well-structured and written to the point. The data are of good quality and the statistics sound. I consider the manuscript publishable after minor revision.

Specific comments: While I am not familiar with the technique used for the MgCO3 concentration determination I feel the whole approach would benifit from additional analyses providing spatially resolved data on Mg distribution (e.g. EMPA, LA-ICPMS,

[Figure]

SIMS). I am aware this might not be practical for the current work but would like to encourage the authors to extend the work into that direction. Without such information we can hardly evaluate any effects (e.g. seasonal changes, internal and interindividual variation) which until now could be hidden by pooling the sample amount needed for the respective analyses. In the discussion I do miss details regarding the role of photosynthesis which could compensate for the changes in pH caused by the pCO2 treatment. Respiration, photosynthesis and calcification rate data would be extremely helpful. I finally miss any reference to some recent papers (e.g. Ragazzola et al. 2013 and 2016). Those papers should be included into the discussion as they provide direct evidence for the CCA's response to OA (while carried out using a different species from a different habitat) on the microscale, chemically and structurally. As this topic directly relates to the use of Mg in CCA as environmental proxy for paleo temperature reconstruction the discussion could extend further into that particular direction. This is meant twofold: 1) with respect to the primary incorporation of Mg and 2) the preservation of the proxy signal in the fossil record.

technical comments: L78: (3) instead of 3/ L226: Rees et al. 2005 missing in reference list

---

## Referee Comment (RC3) · Anonymous Referee #3 · 20 Jul 2016

This study tests for alteration of skeletal minerology in a Mediterranean crustose coralline alga cultured under elevated temperature and $pCO_2$ conditions. The authors measured the Mg content using XRD in skeletal materials cultured under four treatments: (1) ambient temperature and $pCO_2$, (2) elevated temperature and ambient $pCO_2$, (3) ambient temperature and elevated $pCO_2$, and (4) elevated temperature and $pCO_2$. They find that temperature but not $pCO_2$ influences Mg incorporation into the skeleton. While the temperature influence on Mg in alga skeleton is a well-established relationship, the absence of a $pCO_2$ influence on Mg incorporation indicates that this species of alga may be able to buffer against future ocean acidification. While this is

an important outcome, I find the discussion and resulting conclusions extrapolating beyond what can be inferred from the results here. In some places, it would be relatively straightforward for the authors to provide additional evidence to support or refute their discussion points. I am also concerned that there was no independent verification of material growth in culture to ensure that only cultured skeleton was analyzed for the current study. Based on these two points, plus a few minor specific comments below, I recommend revisions to this manuscript prior to publication.

Specific comments

L76: an increase in Mg content under increased temperatures is well established in the literature and does not need to be tested here

L78: typo

L103: I am not familiar with the gross growth morphology of this species of alga. It is difficult to understand the difference between the new crust and the surficial crust. I appreciate that the authors included visualization in figure one of the different crusts but I would suggest they include a cartoon or sketch to better show the different crusts and how they differ.

L105: Is there any evidence that this new material was grown during the experiment? Was growth not measured or a skeletal stain used to mark new growth? Much of the results depend on this presumption and I think evidence to support is needed.

L 109: Again, please confirm that material sampled for XRD was grown under cultured conditions

L130-156 Present results here in the same order as presented in the methods L103 – 107

L171 – 175: It would be useful if this discussion was expanded to be consistent with depth of discussion elsewhere, or the differences between these crusts types and calcification processes better introduced earlier

177: Analyses of the pre-existing thalli (main thalli) provides a baseline Mg content only if analyzed prior to the experiment, otherwise, analysis of the thalli provides baseline Mg content for experimental conditions.

L209: What does NBS and NCC stand for?

L210: is the second sig fig significant here?

L212 – 214 this could be tested using SEM

―――――――――――――――――――

---

## Editor Comment (EC1) · L.J. de Nooijer (Editor) · 1 Aug 2016

Dear Drs Nash, Martin and Gatusso,

I have carefully read the two versions of your manuscript, the comments provided by the three reviewers and your response to them. The reviewers raised valuable comments that helped improving your manuscript. However, there are some issues raised by the reviewers that still need to be accounted for (in more detail) before potential acceptance in Biogeosciences. I am looking forward to see a final version of your manuscript!

Lennart de Nooijer

[Figure]

Major issues

1. Related to the first concern of Reviewer #2, using XRD may not be a very widely accepted, standard method to accurately determine %MgCO3 in carbonates. The method followed here is based on the one outlined in a previous article by the first author (Nash et al., 2013, J Sed Res 83: 1084-1098). Going through this paper, it doesn't appear as if the obtained %MgCO3 from the CCA samples were (directly) compared to those measured by e.g. solution-ICP-MS. There is also very little information on the used reference material ('a dolomite reference standard', page 1088 of Nash et al., 2013) and the justification for a one-point calibration strategy. Isn't it necessary to correct for e.g. drift during the different runs?

I think the authors need to provide more detail on the analytical strategy followed here. Lines 127 and 128 mention the reproducibility (+/- 0.11 mol%), but not on which material this was based on. Dolomite reference material? If so, what is the mol% Mg of this material and is it similar to that of the CCA's? It also remains unclear how the fluorite was used as an internal standard. Was the F distributed homogenously throughout the powder? How much material was analysed per run, etc. Please provide such basic analytical details to enable a detailed comparison of these settings and the obtained data with those of future studies.

2. Reviewer #2 also mentioned the potential offset in ambient conditions and those in the direct surroundings of the CCA due to e.g. photosynthesis and respiration. Although the added paragraph does refer to (recent) studies on the potential physiological controls that determine the uptake of Mg, there is no mention of the potential mismatch between ambient and micro-environmental conditions. Please add this to the added paragraph. 3. The experiment ran for 1 year, in which the T (elevated or not) varied with that of the nearby Mediterranean (Martin and Gatusso, 2009; Martin et al., 2013). The analysis of the %MgCO3 is assumed to represent the average temperature and carbonate chemistry, but this assumes that the CCA's growth was linear. If, and this is not immediately clear from this or the previous papers, this is not the case, but e.g.
growth was confined to certain months, the inferred impact of T and carbonate chemistry on Mg-incorporation (and dissolution) may be misleading. Please elaborate on this or add cautionary notes both in the methods and in the discussion on this matter.

In addition, there are some less important matters that may help you to improve the next version of your manuscript.

Minor issues

1. Line 28: please exchange 'organisms' and 'the' (like in line 47).

2. Line 29: please remove the comma.

3. Line 32 and throughout the rest of the manuscript: often the word 'mineralogical' is used, whereas only the Mg-content of the calcite is analysed and discussed. The latter suggests reporting of e.g. the mineral phase and such, which may better be avoided.

4. Line 35: insert 'their' after 'buffer'.

5. Line 36: insert 'thereby' before 'enabling'.

6. Line 48: replace 'are formed' by 'consist'.

7. Line 50: the 'calcite' here probably refers to low-Mg calcite (<8 mol%).

8. Lines 59-60: '...which is .... into the skeleton' is already in the first half of this sentence.

9. Line 62: better to use 'Crustose Coralline Algae (CCA)' at the beginning of this paragraph and only use 'CCA' in the rest of the introduction.

10. Line 64: 'differing dissolution conditions' may read better as 'varying saturation states' or something similar.

11. Lines 65-66: '...higher phases of Mg-calcite...' reads better as '...calcite with higher Mg-contents...'

12. Line 67: I don't see how this is a feedback mechanism, please clarify.

13. Line 76-77: I am not a native speaker, but I think that '...12-months exposure...' would read better as '...a 12-month exposure...' or '...12 months of exposure...'.

14. Line 102: consider adding 'in winter' and 'in summer' after '9:15' and '15:9', respectively.

15. Line 125: I guess the Mg content is calculated by also taking into account peak asymmetry (Nash et al., 2013).

16. Line 129: Should be: 'The effects of pCO2 and temperature on mol% MgCO3 were assessed...'

17. Line 135: please add: 'of the CCA's' after 'Mg content'.

18. Line 148: '...on the combination of material from 5 thalli...' reads better as '...on material from 5 thalli combined...'

19. Line 160: spell out 'vs'.

20. Line 173: could the authors include error margins for the value of 0.33 mol increase/ degree?

21. Line 179: informal phrasing: please use something like: 'The Mg content of the pink surficial crust increased with temperature similarly compared to that of the ...' By the way, if there is no statistical testing possible, how do the authors know that Mg increases with temperature for this part of the CCA?

22. Lines 191-192: the temperature range tested here is rather limited (3 ‰C), so that I suggest not to present the reported increase in Mg as a 'robust' temperature response, at least not in general.

23. Lines 199-200: please include Willams et al. (2014, Geochim, Cosmochim Ac 139: 190-204) as a reference for algae with a considerably more sensitive Mg/Ca increase

per degree temperature degree.

24. Lines 224-226: please combine these two sentences.

25. Lines 227-232: this is unclear to me. If the % MgCO3 in the 700T treatment is not significantly different than that of the other treatments, it simply isn't. Comparing with the dissolution chips doesn't change this. Please rephrase.

26. Lines 233-241: please refer to the papers of Ries (2006, Geochim Cosmochim Ac 70: 891-900), Gamboa et al. (2010, J Geophys Res 115) and Krayesky-Self et al. (2016, J Phyc 52: 161-173) as examples of heterogeneity in the Mg content in CCA's.

27. Line 250: could the 'dissolution' also be caused by (micro-)organisms that actively dissolved part of the chips? It looks as if the authors presume in the discussion that any dissolution is a-biological. I wonder if there would be any non-biological dissolution at the saturation states/ MgCO3 contents studied here. Perhaps good to state explicitly what may (or not may) have caused the dissolution somewhere.

28. Line 278: 'less' probably means 'less sensitive'.

29. Line 293: it is obvious anyway that skeletal mineralogy is under biological control, please be more specific.

30. LIne 296: this is relatively well known for many other marine calcifyers (see e.g. Ries et al., 2011, Geochim Cosmochim Ac 75: 4053-4064 for an overview), which may be good to refer to here.

31. Lines 317-321: these sentences don't add much information to the discussion. Consider deleting them.

32. Reference to Hofmann et al. should come after that of Henrich et al.

33. Supplementary information: the tables are not in the same format. Please change.

---

## Author Comment (AC1)

**Mineralogical response of the Mediterranean crustose coralline alga *Lithophyllum cabiochae* to near-future ocean acidification and warming**

Response to reviewers

We thank the reviewers for their positive comments and suggestions. We think the MS has been substantially improved by the consequent edits.

**Reviewer 2**
**R2** *Specific comments: While I am not familiar with the technique used for the MgCO3 concentration determination I feel the whole approach would benifit from additional analyses providing spatially resolved data on Mg distribution (e.g. EMPA, LA-ICPMS, SIMS). I am aware this might not be practical for the current work but would like to encourage the authors to extend the work into that direction. Without such information we can hardly evaluate any effects (e.g. seasonal changes, internal and interindividual variation) which until now could be hidden by pooling the sample amount needed for the respective analyses. I*

**Response**
One of the advantages of powder X-ray diffraction is that all the variation is pooled, so to find significant differences across treatments, despite internal and seasonal variation is an indicator as to the strength of the temperature response. But, we consider these results just a good starting point. The big question is how does Mg change at the cellular scale? As referee #2 suggests we do, we already had planned for further interrogation of these samples to understand changes at the ultrastructure scale. Following referee #2 suggestion we undertook exploratory SEM-EDS to see if more detailed Mg study could be incorporated into a revised MS within a reasonable timeframe. However, as happens so often with the corallines, the first results are more complicated than expected and it will take some time to carry out detailed analyses and interpretation. This work will be well beyond the scope of the present manuscript but will be undertaken in a follow-up study.
No changes made

**Referee #2:** *In the discussion I do miss details regarding the role of photosynthesis which could compensate for the changes in pH caused by the pCO2 treatment.* Respiration, photosynthesis and calcification rate data would be extremely helpful.
**Referee #1** also mentioned metabolic costs.

**Response**
All physiological parameters request were measured and reported in the original papers (Martin et al 2009, 2013) referenced at line 82. This sentence has been edited to note specifically respiration, photosynthesis and calcification rate data reported in Martin et al. (2009 and 2013).

Already in the MS at line 244: 'A biological control of mineralization by coralline algae has already been inferred in *L. cabiochae* because its rate of calcification is maintained or even enhanced under elevated pCO$_2$ (Martin et al., 2013a).

We are hesitant to enter into a discussion about photosynthesis ameliorating for pH or metabolic costs. Using the term 'compensate' infers that there must be a negative influence of CO2 on calcification processes that drives the Mg down and the CCA is responding by actively working to maintain Mg- and that is why there was no change in Mg. The fact that there is no pH-related change in Mg in this experiment does not necessarily mean  Mg would be declining but for the organism doing something to compensate. It may just mean there is no effect. Unfortunately, little is known about the cellular scale controls on Mg uptake, thus speculating about compensatory metabolic responses under differing *p*CO$_2$ treatments is fraught with chasm-sized knowledge gaps. This also applies to referee #2's comment about investigating the spatial distribution of Mg.  We consider even more than that is needed, the beautiful NanoSIMS work of Ragazzola et al. (2016) starts to show the complexity of Mg distribution within a cell wall of just one species.  But without a parallel understanding of the distribution and types of organic molecules within the cell walls, and how these do or do not control Mg content, the ability to reliably extend chemical analytical results to a metabolic context are limited.

We have added a paragraph before the final paragraph addressing the above issues and some from the next comment. See paragraph inserted below next comment.

Referee #2**:** *I finally miss any reference to some recent papers (e.g. Ragazzola et al. 2013 and 2016). Those papers should be included into the discussion as they provide direct evidence for the CCA's response to OA (while carried out using a different species from a different habitat) on the microscale, chemically and structurally.*

**Response**
Ragazzola et al. (2016) came out after this paper was prepared for submission. It is now cited, along with Ragazzola et al. (2013), in the discussion regarding Mg changes. There is already discussion on the Mg and physiological response. As we did not carry out SEM to investigate structural changes or microscale distribution of Mg, we do not have a discussion of this where the Ragazzola structural work could seamlessly fit.  Our hypothesis was to test for changes in Mg content ranther than structural changes. However, we have included Ragazzola's papers and added Hoffman et al. (2012) into the discussion on compensatory mechanisms.

Added as second last paragraph
It is worth considering whether there is a compensatory mechanism enabling Mg-content maintenance in the elevated pCO$_2$ treatment. This consideration implies that the Mg content automatically declines with lower pH (the hypothesis we tested) and the organism must therefore have compensated because the results showed no difference with pCO$_2$.

While there have been many studies on Mg content responses to elevated $_pCO_2$ treatments (e.g. Ries, 2011; Egilsdottir et al., 2013; Kamenos et al., 2013; Diaz-Pulido et al., 2014; Nash et al., 2015; Ragazzola et al., 2016) as yet, there has been no study on the internal cellular-scale metabolic controls on Mg uptake in coralline algae. That is, the controls of Mg uptake are unknown. Is it the internal carbonate chemistry, the type of organic substrate or a combination of both? Without an understanding of the physiological mechanisms that control Mg uptake, it is impossible to do more than speculate about potential compensatory metabolic processes. This inhibition to carrying out an informed analysis of potential metabolic controls on Mg uptake highlights the need for basic scientific investigation into how coralline algae calcify and what role the anatomy and organic substrates play in calcification and Mg uptake. Then, we could start to understand how these processes react to external environmental changes. Ragazzola et al. (2013, 2016) and Hoffman et al. (2012) have shown anatomical changes in response to $pCO_2$ that may be ameliorated over longer time periods but the exact controls on those changes are not known. Here, rather than attributing a complicated compensatory response in a poorly understood cellular scale process, probably the simplest explanation is the most logical, that within this range of $pCO_2$ for the *L. cabiochae* there is no influence of carbonate chemistry on the Mg content of the CCA Mg-calcite and the hypothesis of a $pCO_2$ driven decline in Mg is not supported.

**Referee #2:** *As this topic directly relates to the use of Mg in CCA as environmental proxy for paleo temperature reconstruction the discussion could extend further into that particular direction. This is meant twofold: 1) with respect to the primary incorporation of Mg and 2) the preservation of the proxy signal in the fossil record.*

**Response**- Good suggestion. Thank you. The following two paragraphs have been added in the discussion.

The consistent shift of ~0.33 mol% $MgCO_3$/°C across both the control and $CO_2$ treatments suggests the magnesium change is a robust temperature response and this is of interest for CCA paleo temperature proxies (e.g. Halfar et al. 2000, Kamenos et al. 2008, Hetzinger et al., 2009). A similar ratio was found experimentally in *P. onkodes* [0.37 mol% $MgCO_3$/°C (Diaz-Pulido et al. 2014)]. This ratio is also in agreement with results obtained by XRD of articulate corallines [0.286 – 0.479 mol% $MgCO_3$/°C (Williamson et al. 2014)] and a variety of species [0.36 mol% $MgCO_3$/°C using only XRD results in Chave (1954)] collected across a geographical temperature range. The reports of ratios of up to 2 mol% $MgCO_3$/°C (Halfar et al., 2000: Kamenos et al., 2008, Hetzinger et al., 2009; Caragnano et al., 2014) may be due to different analytical methods or species-specific effects. The similarity of the results obtained using XRD for both experimental and *in situ* corallines supports using a ratio of ~0.3 to 0.4 mol% $MgCO_3$/°C as a paleo thermometer when the analytical methods return an effective spatial average for Mg-calcite and the absence of other carbonates; aragonite, low-Mg-calcite, dolomite and magnesite, has been confirmed.

Prior to utilizing CCA as a temperature proxy, it is necessary to verify that the Mg in the Mg calcite is the primary Mg incorporated during calcification and not the result of diagenesis. The depletion of Mg in the dissolution experiment crusts over eight months indicates this change can be relatively rapid once the organism is no longer protected by living tissue. This is likely to be more of a problem when using fossil branching corallines than thick crusts that retain a living surface layer. Indeed, Kamenos et al. (2008) noted their sub-fossil *Lithothamnion glaciale* had significantly lower Mg in the summer season than their living samples. The likelihood of remineralization was considered by Kamenos et al. but rejected, as remineralization was presumed to be to either low Mg calcite or aragonite. The possibility of remineralization to a lower phase of Mg-calcite, as occurred in this experiment and noted at the base of CCA *Porolithon onkodes* (Nash et al. 2013b) had not previously been reported. The present study experimentally confirms that diagenesis does not necessarily mean a change to aragonite or low Mg-calcite, but instead can be to a lower phase of Mg-calcite thus making it more difficult to detect post-mortem diagenesis from Mg measurements or mineralogy alone. High magnification SEM work would be required to check for remineralization.

**Referee #1**
*The pigmented zone on many corallines does not delimit the living tissue. However, on the thin pieces illustrated in Fig. 1 they may. Nevertheless, the "pink" thallus simply has pigments and it can vary in depth into the thallus. I did not see anywhere, where the thickness of the pigmented tissue was reported for the upper and for the lower surface of these crusts. Given the attention given on mineralization that occurs in the pigmented region, that thickness seems to me to be important. (e.g., see lines 200 – 202)*

**Response**
The following sentences added to methods
The depth of the pink-pigmented crust was ~200- 500 µm but only the surface is sampled by scraping on the top with the aim of collecting predominantly epithallial material. However, we do not refer to it as epithallus because by this sampling method we cannot confirm that no sub-perithallial crust has been included, hence surficial pink crust is the most accurate description of this subsample. Our development of this method has shown that if too much pressure is applied during the scraping then substantial amounts of perithallial crust, that also can be pink-pigmented, may be unintentionally sampled.

**Referee #1** *Line 71: replace "preferentially with differentially.*

**Response** – changed

**Referee #1** *Line 108: Fragments 2 – 3 mm are extremely small with considerable exposed thallus relative to the entire photosynthetic surface area. Could this have had an effect on the outcome of the experiment?*

These fragments grew entirely during the experiment and it is likely the small size did not affect the outcome.  Noting that without a true control, I.e. a piece of similar size grown

over the same duration in the natural environment, it is not possible to confirm or deny a potential size effect due to the experimental conditions. The situation would be different had these been pieces of crust removed from the main crust and placed into the experiment, in which case the greater exposure may be a problem.

No change to MS

**Referee #1** *Line 108: I suggest you replace the second use of the word "diameter" with "thickness".*

**Response** –changed

**Reviewer #3**

**Reviewer #3**
*Specific comments L76: an increase in Mg content under increased temperatures is well established in the literature and does not need to be tested here*

**Response**
The original experiment was designed to test a range of physiological responses to anticipated future changes of rising temperature and $p$CO$_2$, as well as a combination of both. To give the results of the combined treatment context, necessarily a 'control' elevated temperature comparison needs to be made.
We agree with r**eviewer #3** that an increase in Mg is well established, but disagree that this means it no longer needs testing experimentally. As the added discussion on temperature proxies shows, there is a substantial difference in the range of Mg changes in response to temperature across species, i.e. a ten-fold difference ranging from ~0.2 to 2 mol% MgCO$_3$. Consequently there is very much a need to test the Mg response to temperature in controlled experimental conditions across a range of species in order to build a more precise paleo thermometer.
No changes made

**Reviewer #3**
*L78: typo*
Fixed

**Reviewer #3**
*L103: I am not familiar with the gross growth morphology of this species of alga. It is difficult to understand the difference between the new crust and the surficial crust. I appreciate that the authors included visualization in figure one of the different crusts but I would suggest they include a cartoon or sketch to better show the different crusts and how they differ.*
**Response**
Black and white sketch added to figure and caption edited

[Figure]

Figure 1: (a) Top face of the main thallus (MT) of *L. cabiochae* showing the pink surficial crust (PSC) and bottom faces (b) free of crusts at the time of collection and (c) with new crusts grown during the experimental period (photos and drawing S. Martin).

**Reviewer #3**

*L105: Is there any evidence that this new material was grown during the experiment? Was growth not measured or a skeletal stain used to mark new growth? Much of the results depend on this presumption and I think evidence to support is needed.*

**Response**: The surfaces were cleaned of epiphytes at the time of collection and photographed. The new crusts used for analyses were not there when collected from the field and no staining was required to visibly identify the new crust growth.

**Reviewer #**
*L 109: Again, please confirm that material sampled for XRD was grown under cultured conditions*

**Response:** As discussed above and the following sentence added to MS in materials and methods:

> These crusts were confirmed to have grown during the experiment as the pre-existing crust was cleaned and photographed at the time of collection and these growths were not present at that time.

**Reviewer #3**
*L130-156 Present results here in the same order as presented in the methods L103 – 107*
**Response**- The order presented in methods L103-107 has been changed to reflect the order in the results.

**Reviewer #3**
*L171 – 175: It would be useful if this discussion was expanded to be consistent with depth of discussion elsewhere, or the differences between these crusts types and calcification processes better introduced earlier*

**Response** The discussion has been expanded and another reference added. See revised paragraph below.

> The pink surficial crust also trended up with temperature and while no statistical analyses could be carried out, these results are consistent with the increase in Mg measured for pink surficial crust as a function of increasing temperature reported in previous work (Diaz-Pulido et al., 2014).  The lower Mg content recorded for the pink surficial crust relative to the bulk crust is in agreement with previous studies on CCA *Porolithon onkodes* (Diaz-Pulido et al., 2014; Nash et al., 2015). Sampling of the surface aims to capture predominantly epithallial carbonate. The epithallial cells of corallines are typically different in shape to the perithallial cells, being shorter and flattened or ovoid shape (e.g. Pueschel and Keats 1997). It is not presently known why the epithallus has a lower Mg than the bulk perithallus or if this offset is common to all species.  However, the close agreement of the temperature response for the *P. onkodes* surficial crust of 0.37 mol% $MgCO_3$/°C (Diaz-Pulido et al. 2014) with the 0.33 mol% $MgCO_3$/°C measured for the new crust in this experiment does suggest the controls on Mg uptake are similar for both the perithallial and epithallial cells.

**Reviewer #3** *Analyses of the pre-existing thalli (main thalli) provides a baseline Mg content only if analyzed prior to the experiment, otherwise, analysis of the thalli provides baseline Mg content for experimental conditions.*

**Response**
As XRD is a destructive process, the exact piece cannot be analysed prior to the experiment.  However, it is possible to take subsamples prior to the experiment for analysis and if the mineralogy matches the post-experiment analysis, then this is usually sufficient to establish similarity (e.g. as done in Nash et al. 2013a).  Unfortunately where the focus of the experiment is on biological responses, not mineral, to the frustration of the mineralogist this early subsampling is not usually done. As r**eviewer #3** has noted, this is a problem. We do note in that paragraph that the values for one of the treatments suggest there has been alteration during the experimental duration.

Sentence edited and now reads

> Analyses of the pre-existing thalli (main thalli) provide a baseline Mg content for pre-experimental *L. cabiochae* with the assumption that this has not changed during the experiment.

Further, we have added to this paragraph to highlight experimental design problems and solutions.

'Ideally when carrying out experiments where it is planned to analyse the crust, for either mineral or structural changes, it is best if a subsample is taken from of each piece prior to being placed into the experimental tanks. This way it can be established that post-experiment crust features are truly representative of the environmental sample (e.g. Nash et al. 2013a) and have not been altered by virtue of being placed in tanks for the duration. Problematically some CCA exhibit changes in growth unrelated to treatment after being placed in tanks (Nash et al. 2015). Thus best practice would be to keep aside subsamples, particularly where a species has not already been well studied at the cellular scale, so that it can be determined if the control tanks result in growth and mineral composition comparable to in-situ growth.'

**Reviewer #3**
*L209: What does NBS and NCC stand for?*

**Reponse**
NBS – National Bureau of Standards
NCC was a typo and has been removed.

**Reviewer #3**
*L212 – 214 this could be tested using SEM*

**Response**
As already mentioned above, SEM work is ongoing but beyond the scope of this part of the study.

---

## Author Comment (AC4)

[revised manuscript text omitted]

**457**

---

## Author Response (AR1)

We note that the editor has advised the manuscript is suitable to publish after the revisions listed in previous upload in August. No further changes required.